

# Temporal trend of cardiorespiratory endurance in urban Catalan high school students over a 20 year period

Jordi Arboix-Alió[1,2,*], Bernat Buscà[1,*], Enric M. Sebastiani[1], Joan Aguilera-Castells[1], Sergio Marcaida[2], Luis Garcia Eroles[3] and María José Sánchez López[4]

[1] Faculty of Psychology, Education Sciences and Sport Blanquerna, Ramon LLull University, Barcelona, Spain
[2] Department of Physical Education, Escola Sagrada Familia Urgell (Barcelona, Spain), Barcelona, Catalonia, Spain
[3] Department of United Organization Systems Information, Hospital Germans Trias, Badalona, Catalonia, Spain
[4] Medical Library, Hospital Sagrat Cor, Barcelona, Catalonia, Spain
* These authors contributed equally to this work.

## ABSTRACT

**Background:** Physical fitness is considered an important indicator of health in adolescents. However, in recent years several studies in the scientific literature have shown a considerable lower trend and an alarming worsening of the current adolescents' physical condition when comparing with previous decades, especially in urban populations. The aim of the current study was to analyse the temporal trend in cardiorespiratory endurance (CRE) in urban Catalan adolescents over a 20-year period (1999–2019).

**Methods:** A cross-sectional analysis study considering the 20-m Shuttle Run test (SRT) results obtained in the last 20 years was carried out. 1,701 adolescents between 15 and 16 years old (914 boys and 787 girls) were divided into four groups, corresponding to consecutive periods of five years (Group 1: 1999–2004; Group 2: 2005–2009; Group 3: 2010–2014 and Group 4: 2015–2019). ANOVA was used to test the period effect on CRE and post hoc Bonferroni analysis was performed to test pairwise differences between groups ($p < 0.05$).

**Results:** Results showed a significantly lower performance in CRE in both sexes. The percentual negative difference was 0.67%, 9.6% and 7% for boys and 5.06%, 14.97% and 9.41% for girls, when comparing the performance in 20-m Shuttle Run test for the first period, respectively.

**Conclusions:** Results suggest that the physical fitness of Catalan urban adolescents is lower in both sexes when comparing the different analysed periods of time. Therefore, CRE adolescents should be improved in order to help to protect against cardiovascular disease and other health risks in adulthood.

Corresponding author
Bernat Buscà,
bernatbs@blanquerna.url.edu

## INTRODUCTION

Physical fitness is a set of physical and evaluable attributes related to the ability to perform physical exercise, and it also provides an important indicator of health (*Tomkinson et al., 2017a*). Physical fitness can be thought as an integrated measure of most body functions (skeletomuscular, cardiorespiratory, circulatory, psychoneurological and endocrine-metabolic) involved in the performance of the daily physical exercise (*Ortega et al., 2008*).

Low physical exercise among children and adolescents has become a problem because its consequences constitute a risk factor for the health of general population. Moreover, this low physical exercise is closely related to different diseases such as obesity or diabetes (*Héroux et al., 2013*; *Nechuta et al., 2015*). These diseases are negatively impacting national health systems, not only in adults, but also in children and adolescents (*García-Hermoso, Ramírez-Vélez & Saavedra, 2019*). Besides low physical exercise (less than 60 min per day according to the World Health Organization (WHO) recommendations), adolescents' intake of energy-dense diets and the sedentary lifestyle (sitting time, new technologies abuse or means of transport uses) contribute negatively in health and quality of life. In this vein, the PASOS (Physical Activity, Sedentarism and Obesity in Spanish Youth) study showed that only 36.7% of children and adolescents fulfill the WHO recommendations (*Gómez et al., 2019*). Also, this study showed that the percentage of inactivity was higher in girls in comparison to boys (70.1% vs. 56.1%) and higher in adolescents in comparison to children (69.9% vs. 56.1%). Thus, epidemiologists have been suggested that physical fitness level may play a crucial role in treating overweight and obesity in this population (*Watts et al., 2005*).

One of the most important body functions is cardiorespiratory endurance (CRE), also known as cardiovascular fitness or aerobic fitness, which refers to the ability of the heart, lungs and circulatory system to supply oxygen to functioning muscles for prolonged periods of time (*Armstrong, Tomkinson & Ekelund, 2011*; *Tomkinson et al., 2017b*). Therefore, the CRE physical condition is an important factor that reflects the health condition of the population (*Castillo-Garzón et al., 2006*; *Gualteros et al., 2015*). Concretely, significant associations have been found between CRE and obesity, diastolic and systolic blood pressure, cholesterol levels and cardiovascular health (*Ortega et al., 2008*). Moreover, a systematic physical exercise entails several health benefits for the active population, also affecting positively their self-esteem or social relationships (*Héroux et al., 2013*; *Lang et al., 2018a*; *Ortega et al., 2008*; *Smith et al., 2014*).

It is well supported that many adult chronic health diseases have their origin in childhood, especially in adolescence (*Lang et al., 2018a*). For instance, a recent longitudinal study found a significant association between a low cardiorespiratory fitness level, assessed with 20 m-shuttle run test (SRT), and future cardiovascular disease in Spanish children aged 6–10-year-old. The children with higher cardiorespiratory fitness level were more likely to have a cardiovascular disease risk (boys odd ratio of 7.117 and 4.439 for girls) (*Castro-Piñero et al., 2017*). Other studies have demonstrated that both biological and behavioural risk factors influences the health status in adulthood. Thus, CRE levels in adolescence are moderately to strongly associated with CRE levels in adulthood, therefore

enabling future health predictions (*Lang et al., 2018b*). Moreover, adolescence is a crucial stage towards a healthy lifestyle because puberty is a key period for skeletal mineralization and obesity prevention (*Smith et al., 2014*). In addition, recent studies have shown an increasing trend of adolescents towards a sedentary lifestyle, considering it as the disease of the 21st century (*Mendoza-Muñoz et al., 2020*), and suggesting that the recommended levels of physical exercise are not achieved (*García-Hermoso, Ramírez-Vélez & Saavedra, 2019*).

The comparison of current adolescent physical fitness levels with those of previous decades clearly highlights the foreseen trends (*Tomkinson et al., 2003*; *Westerstahl et al., 2003*; *Suris et al., 2006*; *Tomkinson & Olds, 2007*; *Ferrari, Matsudo & Fisberg, 2015*). In this context, *Tomkinson & Olds (2007)* reported a decline of aerobic fitness average of 3.6% per decade in 25.4 million individuals aged 6–19 years from 27 countries between 1958 and 2003. Likewise, *Tomkinson et al. (2003)* analyzed the 20 m SRT results of 129,882 children and adolescents aged 6–19 years from 11 countries indicating a decline of 4.3% per decade between 1981 and 2000. Although the vast majority of trend studies show a similar behaviour of CRE among adolescents, *Olds et al. (2006)* have shown different slopes according to geographic area, thus suggesting the importance of researches in specific local populations. Indeed, the evolution of CRE in a concrete region or population is crucial to understand the impact of the nutritional, social or healthy habits, together with their physical condition.

In Spain, the data of the AVENA (Nutrition and Assessment of Nutritional Status of Spanish Adolescents) study noted that Spanish adolescents had worse physical condition than other countries (*Ortega et al., 2005*; *Moliner-urdiales et al., 2010*; *Moreno et al., 2006*). However, to the best of our knowledge, there is a paucity of studies analyzing the trends of CRE in Spanish and Catalan adolescents. Therefore, the aim of the present study was to analyse the temporal trend of CRE in a sample of urban Catalan adolescents over a 20-year period (1999–2019). It was hypothesized that adolescent population would show lower performance in the CRE test over the analysed time periods.

## MATERIALS AND METHODS

### Design

This is a cross-sectional study examining the temporal CRE trend of adolescents in the last 20 years. 1,701 adolescents (914 boys and 787 girls), aged 15 and 16 years, were evaluated in 20-m SRT. Tests were conducted by physical education teachers as part of several schools' physical education programs from 1999 to 2019 in Barcelona. A sample size of 1,529 subjects was estimated by using EpiData 3.1 software (EpiData Association, Odense, Denmark), considering a confidence level of 95% and 2.5% accuracy. Because we intended to maintain that accuracy in both sexes, the minimum sample size was estimated in 1,635 students. Sample was selected through a suitable sampling group from several schools in Barcelona metropolitan area. According to the criterion of home proximity that governs the admission process of students in schools, their socio-economic level was generally considered medium-high. The inclusion criteria were age between 15 and 16 years old and attend school regularly (>80% attendance). Students who had a

health problem that could bias any result or prevent you from taking a test of the study and students with high absenteeism (≥20% absences) were excluded from the study. No anthropometric data was considered due to the lack of uniformity in the data and the disparity of measurement instruments.

To compare the results over time, the sample has been disaggregated by sex and divided into four groups, corresponding to consecutive periods of five years. Thus, Group 1 ($n$ = 450) included the 1999–2004 classes; Group 2 ($n$ = 453), the 2005–2009; Group 3 ($n$ = 406), the 2010–2014 and Group 4 ($n$ = 392) the 2015–2019. The distribution of the four-time periods was made arbitrarily in order to have a similar number of subjects in each period to facilitate comparisons.

All protocols conducted in this research complied with the requirements specified in the Declaration of Helsinki (revised in Fortaleza, Brazil, 2013) and all the participants signed informed consents after receiving a clear explanation. The Clinical Research Ethics Committee of Sagrat Cor Hospital (Barcelona, Spain) approved this study with reference number L-GENZ-E 004.

### CRE test

Cardiorespiratory endurance was assessed using the 20-m Shuttle Run (20 m SRT). The 20mSRT is a running test used to estimate an athlete's aerobic capacity ($VO_2$ max). It consists of one-minute stages of continuous, incremental running speed. The initial speed is 8.5 km/h, and increases by 0.5 km/h per minute (*Léger et al., 1988*). The participant is required to run between two lines 20-m apart while keeping the pace with a timed beep. The test ends when the individual fails in maintaining the pace (*Lang et al., 2018a*). The 20 m SRT is simple, easy to administer, and is part of the ALPHA fitness test battery for children and adolescents (*Castro-Piñero et al., 2010*; *Ruiz et al., 2011*). Moreover, the test demonstrates strong test-retest reliability and moderate to strong validity, being considered the best and most popular field-based measurement of CRE among youth (*Mayorga-Vega, Aguilar-Soto & Viciana, 2015*).

### Statistical analysis

Statistical analyses were performed using SPSS (Version 20 for Windows; SPSS Inc., Chicago, IL, USA). The Kolmogorov–Smirnov test was used to check the normality of the data. Descriptive statistics were used to describe the general demographic and practice characteristics of the sample population: means, SDs, and range for continuous variables, and absolute and relative frequencies for categorical variables. The differences in physical fitness (dependent variable) among the groups (independent variable) were tested using one-way-ANOVA. Bonferroni post-hoc tests were used to observe the pairwise differences between groups. The significance level was set at $p < 0.05$ for all statistical analyses. Moreover, effects sizes were reported as partial eta-squared ($\eta_p^2$), with cut-off values of 0.01–0.05, 0.06–0.13 and >0.14 for small, medium and large effects, respectively (*Cohen, 1988*). For pairwise comparison, the Cohen's $d$ effect size was calculated (*Cohen, 1988*), and the magnitude of the effect size was interpreted as <0.2 = trivial; 0.2–0.6 = small; 0.6–1.2 = moderate; 1.2–2.0 = large; >2.0 = very large (*Hopkins et al., 2009*).

**Table 1 Descriptive analysis of results in 20-m Shuttle Run test over a 20-year period. The data are shown as mean ± SD.**

| | Group 1:<br>1999–2004 ($n$ = 450) | Group 2:<br>2005–2009 ($n$ = 453 ) | Group 3:<br>2010–2014 ($n$ = 406 ) | Group 4:<br>2015–2019 ($n$ = 392) |
|---|---|---|---|---|
| Boys ($n$ = 914) | 10.42 ± 2.18 | 10.35 ± 2.27 | 9.42 ± 2.11[*†] | 9.61 ± 1.88[*†] |
| Girls ($n$ = 787) | 6.91 ± 1.72 | 6.56 ± 1.85 | 5.91 ± 1.66[§‡] | 6.18 ± 1.67[§] |

Notes:
[*] Statistically different than Group 1.
[†] Statistically different than Group 2.
[§] Statistically different than Group 1.
[‡] Statistically different than Group 2.

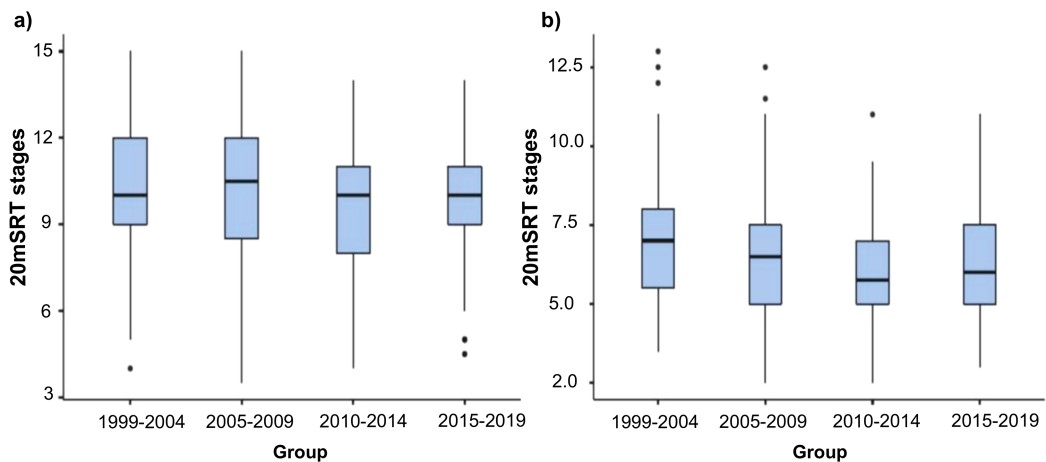

**Figure 1 Comparison of 20 m Shutlle Run Test results between the groups. (A) Boys, (B) girls.**

## RESULTS

Results showed a significantly lower CRE performance in both sexes when comparing the different temporal groups ($p < 0.05$). Boys groups showed a lower performance in the CRE physical capacity, which was significant from the period 2005–2009 onwards ($F_{(3,911)}$ = 13.67, $p$ = 0.000 , $\eta_p^2$ = 0.043). Group 1 (1999–2004) reported higher performance (10.48 ± 2.18) as compared to the Group 3 (2010–2014) and to the Group 4 (2015–2019) (9.42 ± 2.11, $p$ = 0.000, $d$ = 0.49; 9.61 ± 1.88, $p$ = 0.001, $d$ = 0.43; respectively). Similarly, Group 2 (2005–2009) also reported a higher performance (10.35 ± 2.27) compared to 2010–2014 and 2015–2019 groups (9.42 ± 2.11, $p$ = 0.001, $d$ = 0.42; 9.69 ± 1.98, $p$ = 0.046, $d$ = 0.36; respectively), showing lower CRE. Additionally, Group 4 (2015–2019) showed a non-significant upward trend in comparison with the group 3. Table 1 and Fig. 1 summarize the results of the 20 m SRT for each group for both sexes.

The girls trend showed a similar pattern. There was a significantly lower CRE over time ($F_{(3, 784)}$ = 11.76, $p$ = 0.000, $\eta_p^2$ = 0.043), and an upward trend, although non-significant, in the 2015–2019 group respect the 2010–2014. Furthermore, girls reported the highest performance in the 1999–2004 group (6.91 ± 1.72 stages), and showed a significantly lower CRE performance in the 2010–2014 and 2015–2019 groups (5.91 ± 1.66, $p$ = 0.000, $d$ = 0.59; 6.18 ± 1.67, $p$ = 0.009, $d$ = 0.43; respectively).

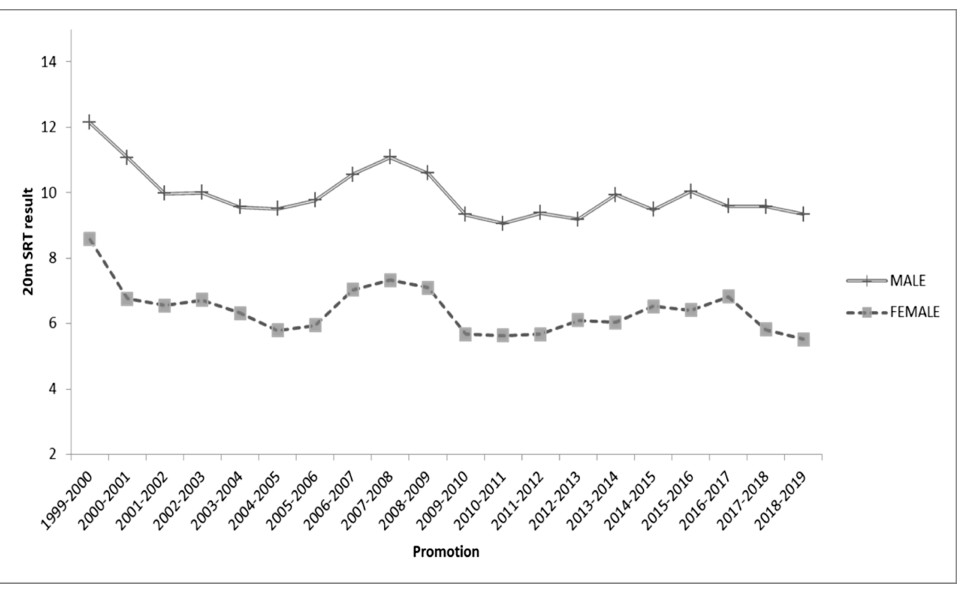

**Figure 2 Evolution of 20 m Shuttle Run Test results over the last 20 years.**

The percentual comparison in boys was 0.67%, 9.6% and 7% chronologically lower respect the 1999–2004 group. On the other hand, girls showed a chronologically lower CRE performance of 5.06%, 14.97% and 9.41% respect the 1999–2004 group (Fig. 1). Figure 2 shows the temporal trend of the 20 m SRT results during each promotion since 1999. For the 20 years analysed, CRE results both for boys and girls followed a similar trend.

## DISCUSSION

The present investigation aimed to analyse the temporal trend of CRE in a sample of Catalan adolescents over a 20-year period (1999–2019). The main finding of the study confirms a significantly lower performance of the most recent groups (3 and 4) compared to previous groups (1 and 2) in CRE, assessed with 20 m SRT, in adolescents of both sexes in the last 20 years.

These results are in line with those previously reported, showing a general and progressive downward trend in adolescents' aerobic capacity compared to those of the previous decades (*Arboix-Alió et al., 2014*; *Ferrari, Matsudo & Fisberg, 2015*; *Suris et al., 2006*; *Tomkinson et al., 2003*; *Westerstahl et al., 2003*). Concretely, in the present study successive groups achieved a lower percentual CRE performance of 0.67%, 9.6% and 7% in boys and 5.06%, 14.97% and 9.41% in girls when compared with the first group (1999–2004). Similar to the findings of the present study, *Tomkinson & Olds (2007)* and *Tomkinson et al. (2003)* reported lower CRE performance around 4% per decade in subjects aged 6–19 years of different countries. In Spain, the data of the AVENA Study (Nutrition and Assessment of Nutritional Status of Spanish Adolescents) reported that adolescents had worse physical condition respect other countries, and estimated that one out of five adolescents were at risk for future cardiovascular events in adulthood

(*Ortega et al., 2005*). Apart from the physiological benefits of physical exercise (*Wu et al., 2019*), it has also been reported its benefits on mental health (*Asare, 2015*), bone health (*Bland et al., 2020*) or cognitive abilities, and how it can help children in the learning processes. Indeed, an improvement of aerobic capacity increases monoamines (dopamine, epinephrine and norepinephrine), resulting in short- and long-term changes in the structure and functioning of brain regions that are responsible of learning (*Best, 2010*), promotes angiogenesis and neurogenesis in the hippocampus, which is the part of the brain responsible of memory (*Hassevoort et al., 2016*), and has the potential to induce vascularization and neural growth of some brain regions (*Donnelly et al., 2016*; *Esteban-Cornejo et al., 2017*). Additionally, from a psychological and sociological perspective, schoolchildren with higher aerobic capacity tend to be physically more active, present less sedentary behaviour patterns and spend their free time in activities with greater cognitive involvement (*Rosa-Guillamón, Garcia-Canto & Carrillo López, 2019*).

Despite not having a single factor explaining the trend of lower performance in CRE reported in the present study, it could be speculated that is probably caused by the combination of environmental, social, behavioural, physical, psychosocial and physiological factors (*Garland et al., 2011*; *Ramos et al., 2016*; *Tomkinson & Olds, 2007*). In this vein, physiological changes are affected by physical changes, such as the increase of fat, directly connected with CRE achievements (*Tomkinson et al., 2017b*). It is well reported that additional fat tissue increases the energy expenditure and the oxygen's cost of running. Therefore, this fact would reduce running performance for any given absolute aerobic power value (*Cureton et al., 1978*). Indeed, in many countries, the changes in fatness have coincided with changes in CRE performance (*Dollman et al., 1999*; *Tomkinson et al., 2012*). Consequently, obesity has become a major public health concern and a significant threat to health. The prevalence of overweight and obesity among children and adolescents has increased in all countries and has already reached alarming levels, especially in industrialized countries (*Kosti & Panagiotakos, 2006*). For instance, Spain is one of the countries with the highest incidence of overweight and obesity in the world (*Wijnhoven et al., 2014*). The Spanish National Health Survey (SNHS), reporting data of children and adolescents aged between 2 and 17 years, determined that the percentage of overweight (body mass index) and obesity in Spain was 28.6% (*Mendoza-Muñoz et al., 2020*). For this reason, it has become one of the greatest challenges for public health in the 21st century because it represents a high cost for the healthcare system (*Ortega, Ruiz & Castillo, 2013*). The health consequences of being obese include physiological disorders; such as dyslipidemia, diabetes, and uterine, colon, breast and prostate cancer; as well as to psychological and social disorders. Low self-esteem, feelings of inferiority, lack of control over impulses, depression, antisocial attitudes or inactivity are connected with children and adolescents with overweight (*Bastos et al., 2005*).

Recent research has linked these changes in body composition to eating and physical exercise patterns. The current trend of having prepared and processed foods easily accessible and in larger portion sizes, together with insufficient levels of physical exercise, are closely associated with social and economic problems (*Ortega et al., 2008*). Although *Bastos et al. (2005)* pointed out that there is no one single factor inducing the development

of obesity (genetics, nutrition, psychology, social and physical inactivity), showed the inactivity as the most relevant, considering that physical exercise plays a vital role in maintaining a healthy lifestyle. The present study analysed the CRE trend of urban population of med-high socio-economic level. This population is generally involved in physical exercise in the school and/or family contexts. For this reason, one can reasonably think that adolescents with lower socio-economic levels and worse physical exercise engagement, might present a worse downward CRE pattern in these periods (*Serral-Cano et al., 2019*). Concerning the CRE performance, *Artero et al. (2009)* suggested that overweight and obese adolescents achieve lower performances compared with their normal-weight counterparts in all tests requiring propulsion or lifting of the body mass like the 20 m SRT, while *Tomkinson & Olds (2007)* reported a stronger negative relationship between fat mass and distance running. Therefore, it seems reasonable to think that an increase in overweight in adolescent population could be closely related to fitness decrease, and thus lower CRE as reported in the present study.

Another reason connected with physical inactivity, and therefore explaining the lower CRE performance trend showed in the present study, is the use of new technologies. An inappropriate or abusive use can have important negative consequences for children and adolescents. Nowadays, new technologies (social networks, internet, smartphones, video games, and television) have become popular hobbies for children and adolescents. However, these sedentary behaviours detract from more physically active leisure time pursuits. According to data from the Spanish National Institute of Statistics, 91.8% of children between 10 and 15 years old are regular Internet users. Likewise, estimations from different studies suggest that a large number of young people in developed countries spend more than 4 h per day watching TV, twice the recommended maximum time (*Lavielle-Sotomayor et al., 2014*). A study from the Pfizer Foundation (2009), reported that 98% of Spanish people aged 11–20 are online platform users, and 70% of them access to the Internet more than 1.5 hours per day. Results of the present study showed a remarkable low CRE performance for girls, but not for boys, in the last two years analysed (Fig. 2). The irruption of social networks (Instagram, Tik Tok or Snapchat) among female adolescents in the last years is a social phenomenon that have changed her lifestyle and, therefore impacting their CRE.

Despite not being statistically significant, it should be noted a reverse trend between the 2006–2007 and 2009–2010 promotions, with a slightly better performance in CRE for both sexes. This slight increase could be related to some local sports events, which could promote physical exercise between the citizens like the 2009 Davis Cup or the 2010 European Athletics Championship, both held in Barcelona. Unfortunately, temporal trends in these behaviours are not clear, because of the difficulty in obtaining accurate measurements and temporal differences in sampling and methodology (*Ekelund, Tomkinson & Armstrong, 2011*). Sustained improvements may require changes to promote healthier habits and environments, especially in schools.

In future studies, it would be interesting to increase the sample size. Furthermore, weight, body mass index, nutritional status, or sedentary habits have not been examined; therefore, given the impact on the results of the fitness tests, it would be interesting to

collect and analyze such data. Similarly, future studies should also consider different socioeconomic status of the sample for contextualizing the adolescents' trends in physical fitness. Moreover, differences in testing conditions like climate, practice, or running surfaces and measurement errors (e.g., methodological drift and diurnal variation) could be considered as a limitation.

## CONCLUSIONS

The present study reports a significantly lower CRE performance in a sample of urban Catalan adolescents aged between 15 and 16 years old, for both genders, over a 20-year period. Concretely, girls showed a remarkable downward trend in the last period analysed (2015–2019).

In light of the results obtained in the present study, as well as according to the WHO many adolescents do not reach the recommended demands of weekly physical exercise, school and community-based youth programs should increase efforts to promote physical exercise among children and young people. Additionally, the findings of the present investigation are of interest from a curricular point of view since the existence of CRE downward trend in adolescents could be a strong reason to raise the status of Physical Education, which traditionally, has been perceived by the educational community as a less important subject than other more traditional areas of the curriculum.

### Funding
The authors received no funding for this work.

### Competing Interests
The authors declare that they have no competing interests.

### Author Contributions
- Jordi Arboix-Alió conceived and designed the experiments, performed the experiments, analyzed the data, prepared figures and/or tables, authored or reviewed drafts of the paper, and approved the final draft.
- Bernat Buscà performed the experiments, analyzed the data, prepared figures and/or tables, authored or reviewed drafts of the paper, and approved the final draft.
- Enric M. Sebastiani performed the experiments, authored or reviewed drafts of the paper, and approved the final draft.
- Joan Aguilera-Castells analyzed the data, prepared figures and/or tables, and approved the final draft.
- Sergio Marcaida performed the experiments, authored or reviewed drafts of the paper, and approved the final draft.
- Luis Garcia Eroles conceived and designed the experiments, analyzed the data, authored or reviewed drafts of the paper, and approved the final draft.
- María José Sánchez López conceived and designed the experiments, performed the experiments, authored or reviewed drafts of the paper, and approved the final draft.

## Human Ethics

The following information was supplied relating to ethical approvals (i.e., approving body and any reference numbers):

The Clinical Research Ethics Committee of Sagrat Cor Hospital (Barcelona, Spain) approved this study with reference number L-GENZ-E 004.

## Data Availability

The raw data are available in the Supplementary File.

## Supplemental Information

Supplemental information for this article can be found online at http://dx.doi.org/10.7717/peerj.10365#supplemental-information.

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
