# Peer review of "Temporal trend of cardiorespiratory endurance in urban Catalan high school students over a 20 year period"

_PeerJ, doi:10.7717/peerj.10365_

## Round 0.1 · original submission · Major Revisions

Reviewers acknowledged the importance and practical applications of the study. However, they have detailed a number of issues that should be considered in a revised version of the manuscript.

Reviewer 1 ·

Basic reporting

The manuscript text presents some introduction points that can be improved.The general write structure is correct. My specific suggestions are described in the attached file.

Experimental design

The objectives of the research need to be rewritten according a cross-sectional trend study. The main lack in this research is that authors affirm that is a longitudinal study and they present a cross sectional analysis.

Validity of the findings

Line 191- The main inconsistency of the study is revealed in the statistical analysis. The authors call the research of "longitudinal", however, they use an adequate test for independent samples, that is, groups with different people. This is what seems to happen. Thus, I suggest that the authors change the design of the longitudinal, for a trend study,a comparison of cross-sectional data over 20 years. The dataset confirm this hypothesis.
According this, The way of writing in all results (line 207) is compromised whereas the data of the study point to a comparison of different subjects. Perhaps the study is a mixed cohort. however, it is not longitudinal. I suggest not saying "increase" or "reduction", we must to say the X CRE value was greater or less than year Y. This is because they are not the same subjects, there is no effect of time and neither maturation, and nor physical activity effects in the present sample.

Additional comments

Thank you very much for the opportunity to evaluate your research. It is an interesting study because it has a greater sample of different years, with important changes in cardiorespiratory fitness. However, some methodologic adjustments are needed. I hope the contributions are relevant to improvements in the general presentation of research.

Annotated reviews are not available for download in order to protect the identity of reviewers who chose to remain anonymous.

·

Basic reporting

I think the paper is very interesting and is about very important children's and adolescents' health indicators. In fact, cardiorespiratory fitness is one of the most important physical fitness components that needs to be highlighted over physical education classes and in several sports and activities for the youth population. In my opinion, your paper showed the necessity of several approaches that aim to prevent and treat several diseases by physical activity since early in life.

These results are very important and need to be considered as a public health policy.
Further, I really like the references that you have used, as well as, the manuscript is very well written, congrats.

I suggest some basic review. I hope you can be able to provide:
1. I suggest you to better present your results.
2. I suggest you change your conclusion over the abstract. Please, what do you think you could really suggest considering your main results? I really like your general conclusion, however in my opinion you could highlight more your results.
3. Please could you discuss a little more you results?

Experimental design

1. I suggest you improve the description of the sample selection. How have you chosen these samples of adolescents? Did you invite all schools?

2. Please, could you include a better presentation of your figure and table?

3. Please, could you include a legend bellow the figure and table?

4. Please, I suggest you include more information about your sample characteristics such as weight, height, BMI, mean age. Moreover, if you have some information about the schools. It is very important for us to understand the sample characteristics. Thank you.

Validity of the findings

Please, could you deeply present how you proceeding to select your sample?
It is a representative sample?

I suggest you to included several considerations about the limitation of your study. For example, there are several variables that you could include as a covariate? Is this a representative sample? Despite we can be able to understand what is happening among this population, how is our limitation when we do the inference of your results?

Additional comments

Please, over the background and the discussion topics I suggest you add the importance of the relationship between physical fitness with all other health indicators such as mental health, bone health, cognition, etc.

Again, please could explore the approaches that should be highlighted considering the great necessity of improving youth physical fitness?

Reviewer 3 ·

Basic reporting

This is an interesting topic and has been discussed in the literature. Regarding the manuscript, it was possible to identify some strengths and points to be improved.

Strengths:
- The topic is relevant and evidence is needed.
- There is a practical implication for the results of the present study.

Points to be improved:
- In the background, despite being well referenced, I suggest a reorganization of ideas to better contextualize the theme and present the aim of the study. For example: start talking about physical activity and how it impacts physical fitness, after decreasing physical fitness over the years and how it impacts the health of children and adolescents. Then justify the investigation of the theme and present the objective.

Experimental design

- The objective is presented for the first time in the methods (it is important to add in the background and standardize since the aim presented in the abstract, in the methods, and in the discussion are different).
- On lines 157-158 the term "physical teacher" is correct?
- When you talk about schools, you use the term "several schools". But how was it selected? What about adolescents? Sample for convenience, voluntary? It is necessary to specify.
- It is necessary to include a sample calculation.
- Finally, the major methodological issue is in the design of the study. As I understand it, they are not the same adolescents over time. Thus, it cannot be considered a longitudinal study. I believe it is a red transversal trend.

Validity of the findings

- In the results, it is important to include more information for the sample characterization (and not the population), even if it is not the aim of the study. Just knowing how many boys and how many girls and describing CRF averages, are not enough to describe the sample.
- In the discussion, the authors should be more specific to answer the study aim and address the main results. I find the idea of speculation interesting, but it is important not to shift the focus so far from the results found in the study.
- There are methodological issues that compromise the conclusions. It is necessary to review mainly the study design and to include the sample size calculation.

Additional comments

- I would also suggest that extra information be removed from the figures’ and tables' titles (it could be described in the legend or in the text that advertises the figures/tables).
- With regard to English, it is necessary to revise. Standardize the terms teenagers and adolescents (I suggest for adolescents) and correct the word "behavior"

---

## Round 0.2 · accepted · Accept

Congratulations for meeting the high standard publication of PeerJ.

Reviewer 3 ·

Basic reporting

Thank you once again for the opportunity to revise this manuscript.
The authors were concerned with improving the work through suggestions, which greatly increased the quality of the article.

Experimental design

The authors answered methodological questions.

Validity of the findings

The authors answered the validity of the findings questions.